# An Artificial Vibrissa-Like Sensor for Detection of Flows [note 1]

**DOI:** 10.3390/s19183892

**Published:** 2019-09-10

**Authors:** Moritz Scharff, Philipp Schorr, Tatiana Becker, Christian Resagk, Jorge H. Alencastre Miranda, Carsten Behn

**Affiliations:** 1Technical Mechanics Group, Technische Universität Ilmenau, Max-Planck-Ring 12, 98693 Ilmenau, Germany (P.S.) (T.B.); 2Section of Mechanical Engineering, Pontificial Catholic University of Peru, San Miguel 15088, Lima, Peru; 3Institute of Thermodynamics and Fluid Mechanics, Technische Universität Ilmenau, 98693 Ilmenau, Germany; 4Faculty of Mechanical Engineering, Schmalkalden University of Applied Sciences, 98574 Schmalkalden, Germany

**Keywords:** vibrissa, flow sensing, bio-inspired sensor, drag reduction

## Abstract

In nature, there are several examples of sophisticated sensory systems to sense flows, e.g., the vibrissae of mammals. Seals can detect the flow of their prey, and rats are able to perceive the flow of surrounding air. The vibrissae are arranged around muzzle of an animal. A vibrissa consists of two major components: a shaft (infector) and a follicle–sinus complex (receptor), whereby the base of the shaft is supported by the follicle-sinus complex. The vibrissa shaft collects and transmits stimuli, e.g., flows, while the follicle-sinus complex transduces them for further processing. Beside detecting flows, the animals can also recognize the size of an object or determine the surface texture. Here, the combination of these functionalities in a single sensory system serves as paragon for artificial tactile sensors. The detection of flows becomes important regarding the measurement of flow characteristics, e.g., velocity, as well as the influence of the sensor during the scanning of objects. These aspects are closely related to each other, but, how can the characteristics of flow be represented by the signals at the base of a vibrissa shaft or by an artificial vibrissa-like sensor respectively? In this work, the structure of a natural vibrissa shaft is simplified to a slender, cylindrical/tapered elastic beam. The model is analyzed in simulation and experiment in order to identify the necessary observables to evaluate flows based on the quasi-static large deflection of the sensor shaft inside a steady, non-uniform, laminar, in-compressible flow.

## 1. Introduction

The sensing of flows is a common task in industry and science. There are lots of conventional measurement principles like a Prandtl probe or drag force probe sensors [1]. In nature, animals use different mechanisms to detect flows as well. For example, there are the lateral lines of sharks, the trichobothria of spiders or the vibrissae of mammals [2]. The different mechanisms have different advantages and disadvantages. Vibrissae are used for the recognition of object contours or to identify features of a surface texture [3,4]. The all-round nature of these hairs and their sensitivity to different kinds of stimuli yield a powerful sensor. The natural paragon is already adapted to artificial sensor systems to scan objects or to autonomous robots for navigating in an unknown environment [5]. Beside information about objects in the surrounding environment, also meteorological information, e.g., wind speed, can be of interest in this context. Furthermore, perturbations caused by wind can adulterate the signal while scanning an object. Those disturbances must be detected to correct the measured signals. An artificial sensor with multiple functionalities including detection of an object contour and information of a surface texture and of the surrounding flow can be worthwhile for autonomous robot systems, because different types of information are recorded by only one sensor. In order to do so, the mechanisms of a vibrissa in a flow must be investigated and determined to complete the already known processes.

The present work tackles this point with the goal to identify the signal components caused by flow and to analyze the influence of typical vibrissae properties on this process. It advances the preliminary work [6] with regard to an improved experiment and an enhanced theoretical model. In our experiment, a rigid sensor shaft is used as well as a flexible one for a good comprehension of the sensor behavior and effects. The experiment is performed to verify the theoretical model, so that further analyses of the proposed theoretical model are approved and details of the natural paragon can be considered in simulations.

Between the different species, there are differences in shape and structure of the vibrissae but there are some generally shared properties. The mystacial vibrissae are arranged in two arrays (mystacial pads) on both sides of the animal’s muzzle, see Figure 1. Across the array, the hair shafts differ in length, in the base-tip diameter ratio and in their inherent curvature [7,8], but principally, the hair shafts are slender, conical, inherently curved and consist of a multilayer structure whereby the outside is covered with scales [9]. The base of each hair shaft is surrounded by a follicle—sinus complex (FSC). Among other, an FSC consists of elastic tissues, blood vessels and mechanoreceptors [10]. So, mechanical stimuli are gathered, transmitted by the hair shaft and transduced by the mechanoreceptors into a neuronal signal [11]. The mechanical stimuli are probably represented by forces and moments at the base of the vibrissa shaft or rather in the FSC [12]. Generally, there are two groups of mechanoreceptors. Along with the slow adapting mechanoreceptors, the rapid adapting mechanoreceptors are sensitive for different frequency ranges [13]. Beside these similarities, the shape of the mystacial vibrissae of seals is undulated while the mystacial vibrissae of rats are smooth [14]. Furthermore, the scales on the outside of the vibrissae are of different sizes. The scales of seal vibrissae have an average diameter of 0.1μm [15] and the scales of rats 5μm [9]. Observations show that the seals align their vibrissae perpendicular to the flow but there is no active whisking through the flow [16,17]. Even, rats do not whisk active through the flow [18]. Inside a flow, the vibrissae deform downstream and oscillate around the deformed shape, whereby the bending moment at the FSC increases approximately linearly [18,19]. In the case of seal vibrissae, the undulated shape minimizes these vibrations [20] and furthermore, it yields to a reduction of drag and lift forces [21,22].

Flow-induced vibrations, like vortex-induced vibrations are intensively studied in the context of natural vibrissae as well as in classic fluid mechanics. There are also flow sensors working on these principles, but the present approach focuses on the quasi-static behavior of the sensor inside a flow and, consequently, the amount of discussion regarding dynamical effects is small, but nevertheless, these effects are quite important. A review discussing these effects can be found in [6].

The advantage of an undulated shape is also reported in classic fluid mechanics for a bluff cylinder with, e.g., grooves. Bluff, straight, clamped or elastically supported cylinders and their behavior inside different types of flow are well known in the literature [23], but for flexible cylinders with low bending stiffness, there are less information and the observed effects and relations differ compared to the ones of bluff bodies. For example, investigations on plants inside an air flow report that there is an elastic reconfiguration of flexible structures due to the flow [24,25]. Since bluff bodies are subjected to a cross- or an axial-flow, flexible ones can be subjected to both types of flow at the same time. In this case, the cross-flow causes a large deformation of the structure with the consequence that the cross-flow acts like an axial-flow on parts of the deformed structure. The large deformation of structure coincides with a change of the drag force as well. The projected obstruction area, the drag coefficient and the impact of the flow velocity are changing [24,25,26,27,28]. The magnitude of the deformation of a structure can be estimated by the Cauchy number. The author of [29] purposes a modified form (Equation 2) to take slender structures into account.

Sensors inspired by trichobothria of spiders [30] or cilia [31] as well as artificial MEMS-based sensor concepts [27,32] are excluded from discussion because of their geometric dimension and/or different sensing principles. These systems are often designed to work in the boundary layer of the flow. A summary of these approaches is presented in [33].

The natural paragon is already adapted to artificial sensors, and equivalent sensors without relation to vibrissae are designed. The authors of [34] designed a bluff body with an undulated profile to detect flows in the sea based on vibrations. An array of vibrissae-like sensors is used in [35] to recognize the profile of a perpendicular directed flow. Here, the signals are measured by strain gauges at the base of the sensor shafts. A similar approach is used in [36], measuring the bending moment at the base of the sensors, a tomography algorithm is applied to determine the flow profile. The magnitude and direction of flows can also be detected by the so-called Wake Information Detection and Tracking System [37]. This system consists of a circular arrangement of sensor shafts which are placed in a radial direction. A single sensor shaft is assumed in [38] and only analyzed in simulations. The authors investigate different sizes of the sensor shaft to determine proper working ranges. The deformation of the sensor shafts is assumed to be small. This approach is similar to [39,40]. In [39], the sensor consists of an elastic shaft which is supported by an elastic foundation. This sensor works for slow flow velocities and can detect the viscosity of the fluid, but for faster flow velocities, there are large deviations. The authors explain these deviations with an inappropriate modeling of the physical characteristics. Taking several details of the natural paragon, like the undulated profile, into account, the sensor presented in [40] detects the wake of an upstream placed object by detecting characteristic frequencies.

The above-mentioned information is analyzed and considered in a bio-mechanic model for the sensor; see Section 2. The theoretical model is used to identify the signal components in the experimental data whereby the focus is on the quasi-static part of the signal and the corresponding deformation of the sensor shaft inside the flow; see Section 3. The ability of the flow detection with the present sensor concept is analyzed in Section 3, discussed in Section 4 and some final remarks and future steps are mentioned in Section 5.

## 2. Materials and Methods

The natural vibrissa is analyzed with focus on its behavior inside a flow. Therefore, geometric data for an array of natural vibrissae of a rat (Wistar) is taken from [9] and for structural data from [41]. Since vibrissae have a conical shape, the Reynolds number Re along the hair shaft (diameter *d*) is changing.
(1)$$Re=\frac{v\;d}{\nu }$$
with *v* as fluid velocity and ν as kinematic viscosity. The diameter of a vibrissa shaft is linearly decreasing from base to tip (see Section 1), consequently Re decreases linearly along a vibrissa shaft too. Figure 1a shows a schematic drawing of an array of vibrissae, and part (b) the change of Re across the whole mystacial pad for $v=5\;\frac{m}{s}$.

Across the mystacial pad, the Reynolds number decreases from the lower-left side to the upper-right side.

The state of the art indicates that there is a quasi-static deflection of the vibrissa shaft as well as an oscillation around it. The magnitude of quasi-static deflection can be estimated by the Cauchy number $C_{y}$, see (Equation 2).
(2)$$C_{y}=\frac{\rho_{f}\;v^{2}}{E}\;\vartheta^{3}$$
with $\rho_{f}$ as fluid density, *E* as Young’s modulus and ϑ taper ratio (maximum to minimum cross-sectional dimensions), see Figure 2a. A value larger than one indicates a large deformation inside a flow. For exemplary data, there is no large deflection inside an air flow, but the larger hair shafts inside the array on the left tend to be more deflected by the flow. This holds true only for this particular case; in other cases, there can be large deflections of the hair shaft, like in a water flow.

The possible dynamic coupling of the structure and the flow is estimated by the reduced velocity $v_{R}$ [29].
(3)$$v_{R}=\sqrt{\frac{C_{y}\;\vartheta \;\rho_{v}}{\rho_{f}}}$$
whereby $\rho_{v}$ represents the density of a vibrissa. Figure 2b illustrates that for all vibrissae $v_{R}$ is of the same order of magnitude. According to the author of [29] for $v_{R}\approx 1$, there is a strong dynamic coupling between structure and fluid. The natural frequencies are important for the exemplary data because of the dynamic coupling. The natural frequencies increase from the left to the right across the mystacial pad, see Figure 2c. This is consistent with the findings in [18].

As a consequence of the literature findings in Section 1 and the performed analyses, information abouta flow seems to be included in the quasi-static deformation of the hair shaft and the low frequent part of signals at the FSC, and in the high frequent part of the signals. This hypothesis is supported by the facts that the FSC consists of different types of mechanoreceptors that are sensitive for different frequencies [13] and, as mentioned in Section 1, flow sensing takes place in water as well and, therefore, large deflections of the hair shaft are also possible.

Considering this preliminary analysis of a vibrissa and the mystacial pad, respectively, the artificial sensor is modeled for simulation and designed for the experiment. In the first step, the quasi-static part of the signal is in focus and the dynamic one is neglected. To identify the different effects and relations, the sensor shaft is assumed to be cylindrical and slender. The sensor shaft is one-sided clamped.

### 2.1. Modeling & Simulation

The flow is assumed to be steady, non-uniform, laminar, in-compressible and the no-slip condition is fulfilled. The direction of the flow velocity *v* is parallel to the *x*-direction. Furthermore, the flow affects the structure, but the structure does not affect the flow (one way fluid-structure-interaction). So, the flow velocity profile is not changing due to the structure. Consequently, the model predicts a drag force (force in *x*-direction) only and no lift force (force in the *y*-direction). Moreover, the deformation of the vibrissa is restricted to the *x*-*y*-plane. Therefore, forces acting in the *z*-direction are not taken into account. In a first step, the drag coefficient $c_{D}$ is assumed to be constant over all flow velocities and deformed shapes, see Figure 3. So, the drag reduction is only caused by the change of the projected obstruction area $A_{⊥}$ as consequence of the elastic deformation of a flexible structure loaded by a flow.

The considered vibrissa is assumed as a homogeneous, straight, slender beam with circular cross-section and the deformation is limited to the *x*-*y*-plane. Inspired by the biological tactile hair, the vibrissa is modeled conically. This yields (Equation 4) describing the diameter of the shaft along the arclength *s*, whereby $d_{0}$ represents the diameter at the base and ϑ the taper ratio.
(4)$$d\left(s\right)=\frac{d_{0}}{L}\left(\frac{s}{\vartheta }+L-s\right)$$

The material characteristics are linear and satisfy Hooke’s Law with Young’s Modulus *E*. In general, the sensor is modeled as one-sided clamped beam loaded by a flow. The flow is represented by a flow velocity profile v(y) that is related to the drag force (Equation 5) or (Equation 6), [23]:(5)$$F_{D}=\frac{1}{2}\;\rho_{f}\;c_{D}\;A_{⊥}\;v_{0}^{2}\;;\;\;\left[F_{D}\right]=N$$
or
(6)$$Q_{D}=\frac{1}{2}\;\rho_{f}\;c_{D}\;d\left(s\right)\;v{\left(y\right)}^{2}\;;\;\;\left[Q_{D}\right]=\frac{N}{m}$$
with $c_{D}$ as drag coefficient. The deformation of the vibrissa is evaluated using the non-linear Euler-Bernoulli beam theory and is limited to the x−y− plane. The corresponding equations are formulated in the system of non-linear differential Equation (Equation 7).
(7)$$\left.\begin{array}{cccc} \; & \frac{dq_{x}}{ds}-Q_{D}\sin\left(\varphi \right)\; & \; & =0\\ \; & \frac{dm_{z}}{ds}-q_{x}\sin\left(\varphi \right) & \; & =0\\ \; & \frac{d\varphi }{ds}-\frac{64\;m_{z}}{E\pi {\left[d\left(s\right)\right]}^{4}} & \; & =0\\ \; & \frac{dx}{ds}-\cos\left(\varphi \right) & \; & =0\\ \; & \frac{dy}{ds}-\sin\left(\varphi \right) & \; & =0 \end{array}\;\;\right\}$$

This system of non-linear differential equation is solved respective to the boundary conditions listed in (Equation 8).
(8)$$x\left(0\right)=0;\;y\left(0\right)=0;\;\varphi \left(0\right)=\frac{\pi }{2};\;q_{x}\left(L\right)=0;\;m_{z}\left(L\right)=0$$

The boundary value problem illustrated in (Equation 7) and (Equation 8) is solved numerically.

### 2.2. Experiment

The experiments are done in a closed wind tunnel with an operating range up to $30\;\frac{m}{s}$. The test section is 0.2m wide, 0.3m high and 0.5m long. The design of the wind tunnel is optimized in a manner to guarantee a nearly constant flow velocity within the entire cross section and a low turbulence intensity. The flow velocity is measured with a constant temperature anemometer (DANTEC Dynamics Miniature wire probe, straight: 55P11, calibrated by DANTEC Dynamics StreamLine 90H02 Flow Unit). The flow velocity is controlled by an open-loop strategy. The influence of additional objects inside the test section, e.g., the sensor shaft, is supposed to be marginal and is therefore neglected. The sensor shaft is placed at the middle of the test section. The used sensor shafts and their properties are listed in Table 1. The shafts D4, D5, D6 are assumed to be rigid and D03 is assumed to be flexible. Furthermore, the rigid sensor shafts are machined at their bases to fit into the miniature jaw chuck that is used as clamping, see Figure 4. The initial 0.02m of the sensor shafts enter in the miniature jaw chuck and get clamped, while 0.15m stay free. The periphery, e.g., force sensor, is located outside of the test section during experiments. The sensor shaft passed through a small hole in the bottom plate of the test section, see Figure 4. Consequently, only a length of 0.134m of the sensor shaft is loaded by the flow. All shafts are polished and cleaned to minimize effects of different surface roughness.

The forces at the base of the sensor shaft are measured with a 3D force sensor (ME-Meßsysteme K3D40, accuracy class 0.5, nominal load ±2 N). The sensor signals are amplified by a GSV-1A4 M12/2 (ME-Meßsysteme) amplifier, recorded by a NI PXI 6221 M-Series multifunction data acquisition device, and processed by LabVIEW 2017. For each flow velocity, the signals of the forces $F_{x},F_{y},F_{z}$ are recorded for a time interval of 30s with a frequency of 1kHz. The quasi-static part of the signal is assumed to be the mean value of the recorded time interval. In the case of sensor shaft D03, images of the deformed state are evaluated. To compare simulation and experiment, the support reaction $M_{z}$ is calculated by the cross product between the centroid of $Q_{D}$ in *y*-direction and $F_{x}$. The centroid coordinate of $Q_{D}$ in *y*-direction is assumed to be equal to the half of the current tip position in *y*-direction. In the case of D03, the current tip position is determined by image tracking. The images are taken with a resolution of 4608 × 2592 pixels, whereby one pixel corresponds to around 0.077 mm.

## 3. Results

First, the present sensor concept and the simulation results are proved inside a wind tunnel, see Section 3.1. Afterwards, three application examples are discussed: sensing of water flows inside a pipe, sensing of air flows inside a rectangular duct, and sensing of air flows over a flat surface, see Section 3.2.

### 3.1. Proof of Concept and Simulation

The simulation is described in Section 2.1 and the experimental set-up in Section 2.2. For further theoretical and experimental analyses, the drag coefficients $c_{D}$ of the sensor shaft are determined considering the measured support reaction $F_{x}$ and flow velocity $v_{0}$, see Figure 5, Figure 6a and Figure 7b. For all sensor shafts, $c_{D}$ reaches a nearly constant value with increasing $v_{0}$ as expected. In the case of D4, D5, and D6, the determined values for $c_{D}$ scatter around 0.65, see Figure 5a. In comparison, for D03 larger values ≈0.75 are determined, see Figure 5b. Since D03 is assumed to be flexible, the tip position $\left(x(L),y(L)\right)$ is tracked for $v_{0}=\tilde{v}\frac{m}{s},\;\tilde{v}\in \left[0,20\right]$, with a step size of $2\;\frac{m}{s}$. Using this information, in each state the $A_{⊥}$ is estimated and considered while calculating $c_{D}$.

The support reactions $F_{x}$, $F_{y}$ and $M_{z}$ for D4, D5 and D6 are illustrated in Figure 6. For each sensor shaft, the support reactions are increasing with increasing $v_{0}$. Furthermore, the trend is non-linear. The simulated support reactions match the experimental data. In contrast to the theoretical model, the experimental data contain components for $F_{y}$, see Figure 6. For a larger shaft diameter, larger support reactions are measured.

Figure 7a shows the deformed shapes of D03 in simulation and experiment and qualitatively the flow velocity profile. The deformation of the sensor shaft increases for larger $v_{0}$. There are large tip displacements in *x*-direction and small ones in *y*-direction, compare Figure 5c. There is a conformity between the tracked and simulated data. The conformity gets less with increasing $v_{0}$. The support reactions for D03 are smaller than the ones for D4, D5 and D6 while the sensor shaft diameter is smaller too. The trends of $F_{x}$ and $M_{z}$ are less distinct, see Figure 7b,c. The experimental values scatter around the simulated data, but a trend similar to D4, D5 and D6 can be assumed. The values for $F_{y}$ are near zero, and there is no clear behavior, see Figure 7b.

The drag coefficients shown in Figure 5 and the support reactions reported in the Figure 6 and Figure 7b,c are used to calculate the corresponding $v_{0}$, see Figure 8a. This is done applying the algorithm described in Section 2.1. For small $v_{0}$, the deviations of the calculated values for $v_{calc}$ are larger than the ones for large $v_{0}$, see Figure 8b.

### 3.2. Application Examples

Ensuing from the proof of concept, the theoretical model is used to analyze the influence of the properties of a natural vibrissa on the simulated signals/support reactions under different load conditions. A new parameter set is introduced in Table 2 considering findings of Section 1. The new geometrical parameters are adapted from [9] for a vibrissa of type 3B (Column 3, Row B, see Figure 1). The calculated taper ratio matches with the findings in [43]. The Young’s modulus is adapted from [41]. The three application examples are analyzed in simulation because the development of an artificial sensor shaft that satisfies the properties of the natural paragon is challenging. In addition, the support reactions for a sensor shaft with natural properties are very small or the flow velocity is very large, see Figure 9, Figure 10 and Figure 11. To measure support reactions of this magnitude, a more sensitive force sensor is necessary. Further on, to generate a flow velocity in the order of 60 $\frac{m}{s}$ (Example 3) the used wind tunnel is not powerful enough.

**Example** **1.**
*Water Flow inside a Pipe.*


Inspired by drinking water pipes, a water flow with a parabolic flow profile inside a pipe with circular cross section is assumed, see Table 3. Since the theoretical model is restricted to laminar flows, the geometrical dimensions of the pipe and $v_{0}$ are chosen to satisfy Re<2500, which is commonly related to laminar flows.

Figure 9a shows the qualitative flow profile and the deformed shape of the sensor shaft. In general, the deformation is small. The largest deflection occurs at the tip of the shaft. The support reactions $F_{x}$ and $M_{z}$ are proportional to the flow velocity. However, this relation is nonlinear, see Figure 9b,c.

**Example** **2.**
*Air Flow inside a Rectangular Duct.*


A flow inside a rectangular duct is not parabolic like in a pipe, it becomes stretched in dependence on the geometric dimensions of the cross section of the duct. Here, the flow velocity profile is calculated using the model of [44], see Table 4. The present example is motivated by, e.g., air conditioning systems. The geometric dimensions and $v_{0}$ yield a laminar flow again.

Like in Example 1, the flow induced deflection of the sensor shaft is small, see Figure 10a. Furthermore, the calculated support reactions are in the same order of magnitude, see Figure 10b,c.

**Example** **3.**
*Air Flow over a Flat Surface.*


The third example is adapted from [38], an aircraft is assumed to travel with $60\;\frac{m}{s}$ in a height of 8000m. The sensor is placed on the airfoil. The flow around the airfoil is simplified to a flow over a flat surface. The boundary layer is assumed to be free of turbulences because $Re<5\times 10^{5}$ and considered for modeling the flow velocity profile, see Table 5 and [38].

The assumed conditions for the flow cause a large deflection of the sensor shaft, and again $F_{x}$ decreases while $M_{z}$ increases with increasing $v_{0}$, see Figure 11a–c.
(9)$$v\left(y\right)=\left\{\begin{array}{cc} v_{0}\left(\frac{3}{2}\left(\frac{y}{\delta }\right)-\frac{1}{2}\;{\left(\frac{y}{\delta }\right)}^{3}\right) & y\in \left(0,\delta \right)\\ v_{0} & y\in \left[\delta ,\infty \right) \end{array}\right.$$
(10)$$\delta =4.91\;\sqrt{\frac{\nu \;x_{0}}{v_{0}}}$$

The reported support reactions are larger compared to Examples 1 and 2. For all $v_{0}$, the whole sensor shaft is loaded by the drag force $Q_{D}$ whereby the maximum of $Q_{D}$ is located near the base of the shaft in every case, Figure 11d. After reaching the maximum, $Q_{D}$ converges to zero but is not equal zero for the present data. With increasing $v_{0}$, the upper part the sensor shaft (near the tip) gets aligned to the flow direction and $A_{⊥}$ gets reduced simultaneously. In Figure 11e, first the reduction of $A_{⊥}$ is marginal and gets forced with larger $v_{0}$ afterwards.

## 4. Discussion

The drag coefficients $c_{D}$ determined in Section 2.2 are lower than typical values (0.9–1.2) reported in the literature [23]. This can be caused by inaccuracies while determining $v_{0}$. The wind tunnel is controlled by an open-loop strategy, so it is possible that an object inside the test section causes a pressure difference and, consequently, a difference in the flow velocity which is not captured by the control. All sensor shafts are small compared to the geometric dimensions of the test section, thus, a large pressure change is questionable. This also applies to the influence of the surface roughness of the sensor shafts or the surface roughness of the walls of the test section. For all sensor shafts, the measured support reactions are illustrated in Figure 6a–c and Figure 7a,b. In each case, there is a force component in the *y*-direction whereby negative $F_{y}$ corresponds to a force acting in an opposite direction on the shaft. This can be explained by the presence of a lift force (force in *y*-direction) in addition to the drag force (force in *x*-direction), especially for D03, where the shaft is deflected in flow direction the projected obstruction area $A_{⊥}$ in *y*-direction is increasing. Therefore, the development of a lift force is supported. Also, for the rigid sensor shafts D4, D5 and D6, an equal development of a force in the *y*-direction is reported but there are no deformations of the shafts. For clamped, rigid cylinders in a perpendicular oriented cross-flow in *x*-direction, there are no information about the presence of a quasi-static lift force that acts along the longitudinal axis of the cylinder in the literature, but oscillating force components in the *x*- and *z*-directions caused by vortex shedding are reported and related to the lift force loading a straight cylinder whereby the longitudinal axis is parallel to the *z*-direction, e.g., [45]. This is not the case in the present work. Hence, the authors assume deviations regarding the alignment of the force sensor with respect to the flow direction. For example, a rotation of the force sensor respective to the *z*- axis yields a split of the total drag force in a component in eachof the *x*- and *y*-directions and also a different $A_{⊥}$ results, and rotation respective to the *y*- axis increases a force component in the *z*-direction whereby the force in the *x*-direction decreases. So, considering only the component in the *x*-direction, a smaller value for $c_{D}$ results. For all sensor shafts, there is a strong change of $c_{D}$ from small to high Reynolds numbers, followed by a nearly constant trend. This is in accordance to the literature, e.g., [46]. This change is neglected since a constant value for $c_{D}$ is used in all simulations and data treatment. Consequently, there are larger errors calculating small $v_{0}$; see Figure 8. For D03, the deviations are larger than for the other shafts. The measured support reactions of D03 are smaller than those of D4, D5 and D6 and close to the limits of the used force sensor. Nevertheless, the results confirm the theoretical model and emphasize the requirement to consider the change of the $A_{⊥}$. For a rigid, non-deformable sensor shaft, it is sufficient to determine $F_{x}$ and apply (Equation 5), but for the flexible, deformable shaft, all support reactions must be considered in order to determine the present $A_{⊥}$. With respect to the four analyzed sensor shafts, the model of a vibrissa-like flow sensor is confirmed.

The analyzed Examples 1–3. support this statement; see Section 3.2. Here, the sensor shaft is modeled considering values of a natural vibrissa of type 3B (Column 3, Row B, see Figure 1). The inherent curvature, the undulated shape, the internal layered structure and an influence of the surface roughness (size of scales) are neglected and only one vibrissa is considered. The natural vibrissa is simplified to be a straight, conical cylinder consisting of a linear elastic, homogeneous material. In Example 1, the sensor is placed inside a pipe carrying water. In contrast, in Example 2, there is an air flow inside a rectangular duct. Both examples are related to similar Reynolds numbers. Comparing these findings to Figure 1 and Figure 2a, the validity of (Equation 2) for vibrissa is shown. The statements concerning Figure 2b are not further analyzed because the used theoretical model lacks dynamics and is restricted to quasi-static. In both examples, the decreasing bending stiffness due to the conical shape and increasing load by change of the flow velocity along the height of the shaft should increase the deflection of the tip, but these two effects seem to be counteracted by a decrease of $A_{⊥}$ and finally, there are only small deflections; see Figure 9a and Figure 10a. A flow over a flat surface is assumed in Example 3 This scenario is inspired by a flow over an airfoil of an aircraft. Bats are an additional natural paragon for this example because they have sensory hairs on their wings for flight control [47]. Compared to Examples 1 and 2, there are larger deflections of the sensor shaft. For large $v_{0}$, the tip of the shaft aligns to the flow velocity direction until the tip stays nearly parallel. Therefore, the part that remains nearly parallel to the flow direction stays approximately in axial-flow and not in cross-flow. For even larger $v_{0}$, it seems to be possible that this part of the deflected shaft is maybe drafting behind the less deflected part of the shaft. This elastic deflection/reconfiguration is often related to drag reduction in the literature; see Section 1. The drag reduction can be caused by different effects. Based on the findings regarding Figure 5, $c_{D}$ can be determined in dependence on the present Reynolds number, e.g., in [28,46]$c_{D}$ is reported as a function of the Reynolds number. After verifying this approach, (Equation 7) can be modified regarding this relation to take the drag reduction into account. Furthermore, the overall shape of the shaft is changing from a straight cylinder to a curved one. There is a reduction of $c_{D}$ due to the convex shape of the deflected shaft which is more streamlined than a straight cylinder compared to what is reported in [29]. This effect can be amplified if an inherent curvature of the sensor shaft is considered as in nature. Figure 11d indicates the influence of the conical shape. After reaching the maximal value, the load gets reduced by the reduction of the diameter. Furthermore, $A_{⊥}$ decreases with increasing $v_{0}$, see Figure 11e. Another reduction of drag can be caused by the velocity-drag force relation. In the present work, a 2nd order polynomial relation is assumed but for large deflection inside a flow, this relation can change [24,25,26]. For example, the so-called Vogel Exponent describes this behavior. To take these effects into account, the theoretical model must be improved because using the present model, the influences of these effects cannot be evaluated.

Compared to existing vibrissa-like flow sensors, e.g., [1,34], this new approach includes the elastic deflection of the sensor shaft and the related reduction of drag inside a flow. Is this effort really necessary? For a rigid sensor shaft, $A_{⊥}$ is not being reduced for increasing $v_{0}$, consequently, there is no drag reduction. In contrast, for a flexible sensor, the shaft deforms and $A_{⊥}$ gets reduced. This prevents an overload of the structure. This effect was already highlighted by Steven Vogel in [24]:
In short, faced with a given force, it is cheaper (and probably safer) to accept a fair amount of deformation — better bent than broken.

Additionally, it causes another advantage. Considering the elastic deformation, the sensor is applicable in various scenarios with different environmental conditions as demonstrated by Examples 1–3. The approach of [39] includes an advanced artificial sensor that is flexible etc., but it lacks a valid theoretical model to evaluate the flow velocity successfully. In contrast, the theoretical model in [25] compromises a tapered, slender beam and is applicable for a vibrissa-like sensor but there is no experimental validation and it does not focus on a sensor application. Common flow sensors, e.g., anemometers or differential pressure flow meters are different in a major aspect to the present concept. Next to flow sensing, also tactile exploration seems to be possible with the same vibrissa-like sensor. The proposed theoretical model (modelling assumptions) and experimental setup were already used to scan object shape and texture by tactile scanning, e.g., [48]. Both, tactile exploration and flow sensing, are based on the signals measured at the base of the sensor; therefore, a combination of both functionalities in one sensor is what we aimed for. Referring to Section 1, flow is also mentioned as a source of disturbance while performing tactile exploration. Based on the present findings, at least the influence of an air flow with small $v_{0}$ on the quasi-static part of the measured support reactions seems to be low, but not necessarily neglectable. So, flow must be considered as an additional signal component for data processing of tactile signals to correct the captured support reactions.

Since a natural paragon was adapted and analyzed, are the present results transferable to the natural paragon again? The experiment as well as the simulation are limited and restricted by various factors which are not consistent with biological findings, e.g., the vibrissa shaft was assumed to be straight. So, considerations regarding the natural vibrissa are very questionable. The only point that also applies to the natural example is the fact that flow properties are included in the quasi-static part of the signals inside the FSC. In the literature, it is known that there are rapid and slow adapting mechanoreceptors. Those capture low as well as high frequency stimuli during tactile exploration [49]. Considering this point, it can be supposed that flow detection underlies similar signal processing like tactile exploration because the present work discussed low frequency/quasi-static signals and in the literature, there are findings implying that high frequency/oscillating signals contain information about flow [16].

## 5. Conclusions

This work presented a flow sensor with a flexible shaft as infector and a force/torque sensor as receptor, respectively. In this way, the developed theoretical model includes large deflections of the sensor shaft and allows for the detection of a flow velocity using the support reaction at the base of the sensor shaft only. The theoretical model was validated using sensor shafts of different diameters. The sensor shafts with large diameter and high bending stiffness were used to prove the overall concept in comparison to findings reported in the literature. Using a sensor shaft with small diameter and low bending stiffness, the new approach was analyzed in a wind tunnel. The experimental and simulated data match qualitatively. The overall trends are as expected. For all types of sensor shafts, different flow velocities were calculated. Since this procedure is based on support reactions, the qualitative trend matches. The experimental proof of concept was accompanied by three application examples that were theoretically evaluated. They considered a parameter set inspired by natural vibrissa for the sensor shaft. The application examples pointed out effects and advantages of a flexible and tapered sensor shaft for a flow sensor.

The used theoretical model can be advanced in various ways. The most significant limitation is the restriction to a quasi-static behavior. Taking dynamics into account, more effects will arise. An interesting point is if the flow velocity can be detected by a certain frequency. Since this idea is already applied in other commercial flow sensors, e.g., Vortex Flowmeters, and findings in the context of natural vibrissae confirm it, it has to be analyzed in further investigations. Another restriction of the present model is that the flow influences the sensor shaft, but the sensor shaft does not influence the flow. Related to this point is the question of a lift force occuring. The experiments report a force component in the vertical direction, which must be clarified. The advances in simulation must be accompanied by experiments. Therefore, a multi-component force/torque sensor should be used, e.g., [50], to record the support reactions. Finally, experiments like object contour reconstruction or surface texture detection with an artificial vibrissa-like sensor must be combined with a flow as further stimuli to analyze the superposition of information in the measured signals and how it can be processed.

## Figures and Tables

**Figure 1 sensors-19-03892-f001:**
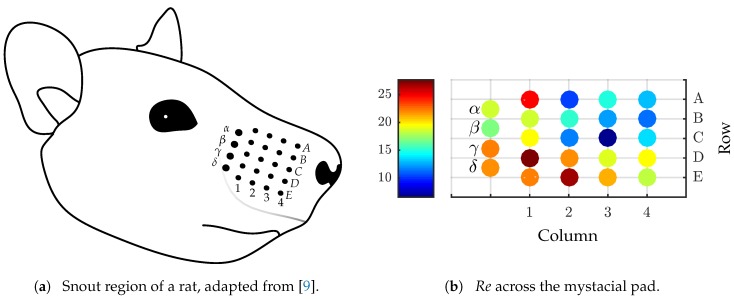
(**a**) Schematic drawing of the mystacial pad of a rat. A black dot corresponds to a single vibrissa, the vibrissae shafts are excluded from the illustration. (**b**) Change of the Reynolds number across the mystacial pad with respect to the mean diameter of each vibrissa and $v=5\;\frac{m}{s}$. Here, the Young’s modulus of the vibrissa is set to 3GPa, the density of the vibrissa $\rho_{v}=1140\;\frac{\mathrm{kg}}{m^{3}}$, the density of air $\rho_{f}=1.2\;\frac{\mathrm{kg}}{m^{3}}$ and the kinematic viscosity $\nu =1.5\times 10^{-5}\;\frac{m^{2}}{s}$ [41,42].

**Figure 2 sensors-19-03892-f002:**
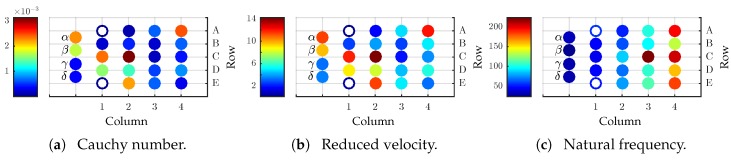
(**a**) Cauchy number and (**b**) reduced velocity in modified form from [29]. (**c**) Natural frequency of the vibrissae across the mystacial pad. (**a**,**b**) consider the same parameters as in Figure 1. Annulus markers correspond to inaccurate data.

**Figure 3 sensors-19-03892-f003:**
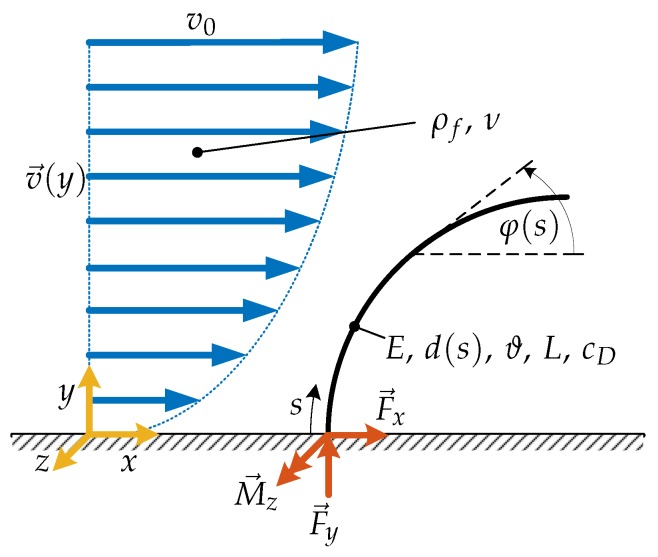
The mechanical/geometrical properties of the beam are described as follows: *E* as Young’s Modulus, d(s) as diameter depending on the arclength *s*, ϑ as ratio of base diameter and tip diameter, *L* as length, $c_{D}$ as drag coefficient and φ as slope to the respective *x*-axis. The support reactions are $F_{x},\;F_{y},\;M_{z}$ and the internal forces are $q_{x}$ and $m_{z}$, see (Equation 7). The flow velocity profile is characterized by v(y) and the maximum flow velocity $v_{0}$ whereby the fluid properties are given by the density $\rho_{f}$ and the kinematic viscosity ν.

**Figure 4 sensors-19-03892-f004:**
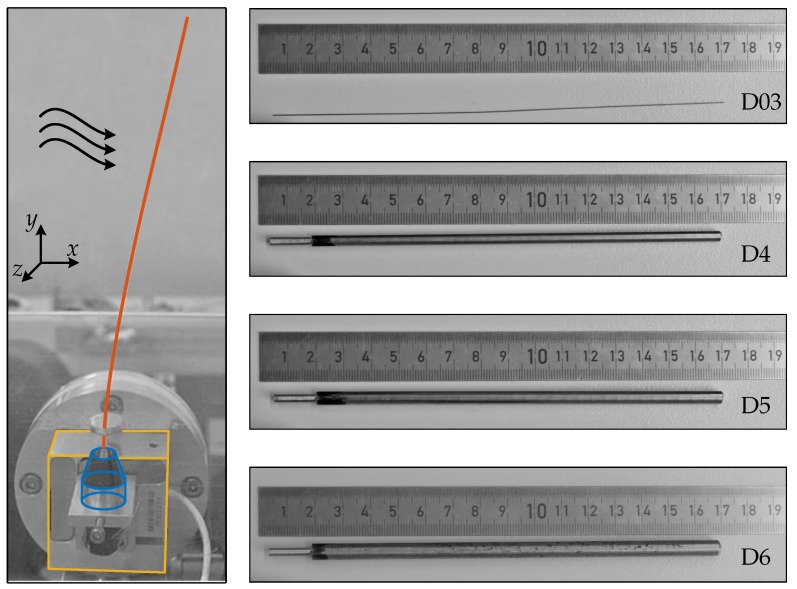
In the **left** picture, the sensor shaft D03 (orange) is clamped by a miniature jaw chuck (blue) and mounted onto the force sensor (yellow) while it stays inside a flow (black arrows). On the **right**, the sensor shafts D03, D4, D5 and D6 are depicted, structural details in Table 1.

**Figure 5 sensors-19-03892-f005:**
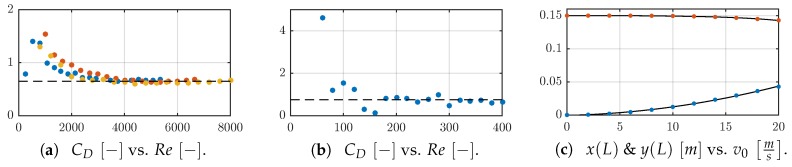
Drag coefficient $c_{D}$ for: (**a**) D4 (blue), D5 (red), D6 (yellow) and (**b**) D03 in dependence on the Reynolds numbers according to $v_{0}=\tilde{v}\frac{m}{s},\;\tilde{v}\in \left[0,20\right]$. The dashed line in (**a**) corresponds to a value of 0.65 and the one in (**b**) to 0.75. (**c**) shows the tip position with x(L) (blue) and y(L) (red) and corresponding curve fits (black solid line) of D03 for $v_{0}=\tilde{v}\frac{m}{s},\;\tilde{v}\in \left[0,20\right]$.

**Figure 6 sensors-19-03892-f006:**
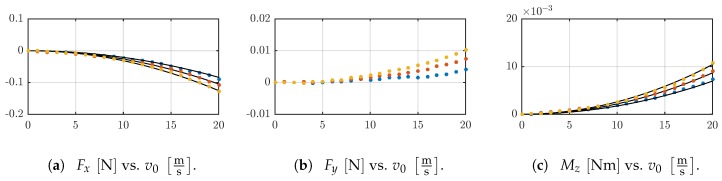
The experimental data sets for D4 (blue), D5 (red) and D6 (yellow) and the simulated data (black solid lines) are illustrated for the support reactions: (**a**) $F_{x}$, (**b**) $F_{y}$ and (**c**) $M_{z}$; in dependence on the maximum velocity $v_{0}$.

**Figure 7 sensors-19-03892-f007:**
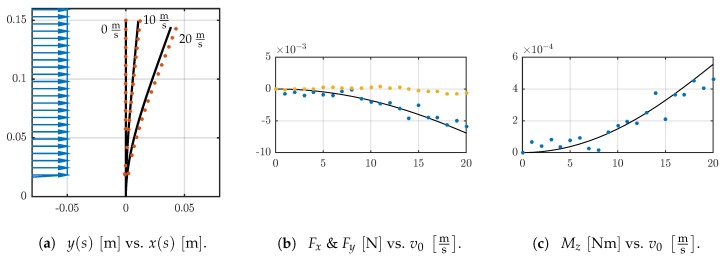
(**a**) The flow velocity profile v(y) (blue) is qualitatively shown for $v_{0}=20\;\frac{m}{s}$ and the deformed shapes of the shaft D03 for $v_{0}=\tilde{v}\frac{m}{s},\;\tilde{v}\in \left\{0,10,20\right\}$ in experiment (red) and simulation (black solid line). Support reactions in experiment (**b**) $F_{x}$ (blue), $F_{y}$ (yellow) and (**c**) $M_{z}$ (blue); each in dependence on the flow velocity $v_{0}$ and corresponding simulations (black solid lines).

**Figure 8 sensors-19-03892-f008:**
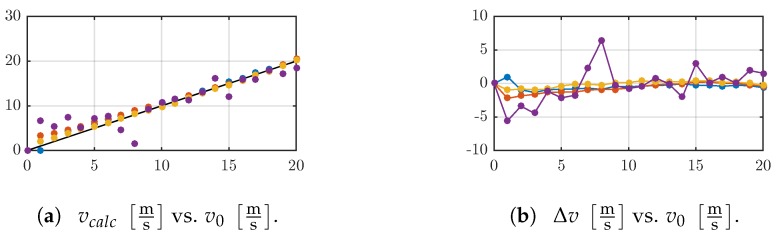
(**a**) The calculated velocity $v_{calc}$ is compared to the measured one $v_{0}$: D03 (magenta), D4 (blue), D5 (red), (D6) yellow. The black line corresponds to the original $v_{0}$. (**b**) According to (**a**), the error Δv is calculated by $\Delta v=v_{0}-v_{calc}$.

**Figure 9 sensors-19-03892-f009:**
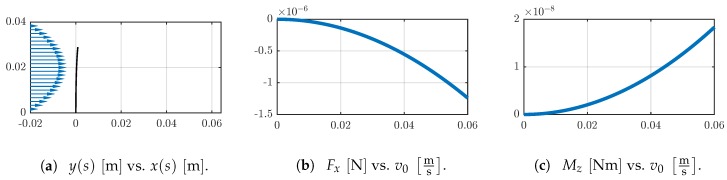
(**a**) The qualitative flow velocity profile v(y) (blue) and the deformed shapes are shown for $v_{0}=0.06\;\frac{m}{s}$. Change of the support reactions (**b**) $F_{x}$ and (**c**) $M_{z}$ in dependence on the flow velocity $v_{0}$.

**Figure 10 sensors-19-03892-f010:**
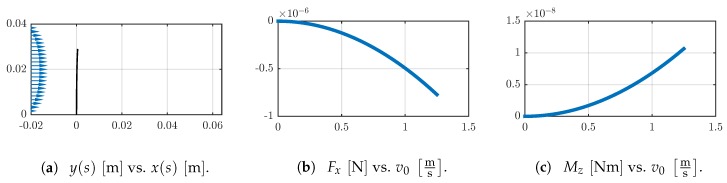
(**a**) The qualitative flow velocity profile v(y) (blue) and the deformed shapes are shown for $v_{0}=1.25\;\frac{m}{s}$. Change of the support reactions (**b**) $F_{x}$ and (**c**) $M_{z}$ in dependence on the flow velocity $v_{0}$.

**Figure 11 sensors-19-03892-f011:**
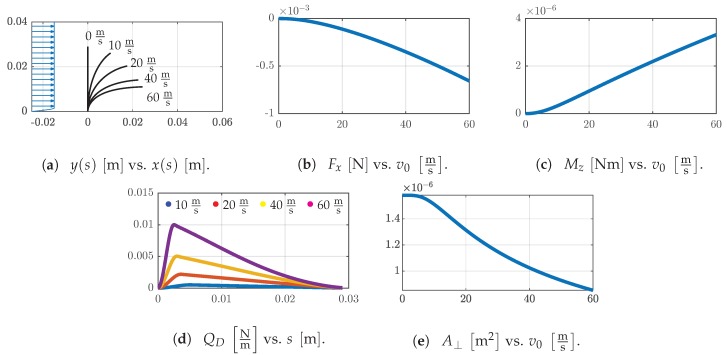
(**a**) The flow velocity profile v(y) (blue) is qualitatively shown for $v_{0}=60\;\frac{m}{s}$ and the deformed shapes for $v_{0}=\tilde{v}\frac{m}{s},\;\tilde{v}\in \left\{0,10,20,40,60\right\}$. Change of the support reactions (**b**) $F_{x}$ and (**c**) $M_{z}$ in dependence on the flow velocity $v_{0}$. (**d**) The drag force $Q_{D}$ along the arclength *s* of the sensor shaft for $v_{0}$ according to (**a**) excluding $v_{0}=0\;\frac{m}{s}$. (**e**) The projected obstruction area $A_{⊥}$ for different $v_{0}$.

**Table 1 sensors-19-03892-t001:** Properties of the used sensor shafts.

Name	*L* [m]	$d_{0}$ [m]	ϑ [-]	*E* $\left[\frac{N}{m^{2}}\right]$
D03	0.15	0.0003	1	$2.1\times 10^{11}$
D4	0.15	0.004	1	$2.1\times 10^{11}$
D5	0.15	0.005	1	$2.1\times 10^{11}$
D6	0.15	0.006	1	$2.1\times 10^{11}$

**Table 2 sensors-19-03892-t002:** Parameters of the sensor shaft for the simulation of the application examples.

$d_{0}\;[m]$	l[m]	$E\;[\frac{N}{m^{2}}]$	ϑ[−]	$C_{D}\;\left[-\right]$
$106\times 10^{-6}$	0.029	$3\times 10^{9}$	≈35	0.9

**Table 3 sensors-19-03892-t003:** Parameter for the simulation of a water flow inside a pipe.

$v\left(y\right)\;[\frac{m}{s}]$	$v_{0}\;[\frac{m}{s}]$	$\rho_{f}\;[\frac{\mathrm{kg}}{m^{3}}]$	$\nu \;[\frac{m^{2}}{s}]$	$d_{p}\;[m]$	Re[−]
$-4\;\frac{v_{0}}{d_{p}^{2}}\;{\left(y-\frac{d_{p}}{2}\right)}^{2}+v_{0}$	0.06	997	$1.003\times 10^{-6}$	0.04	≈2400

**Table 4 sensors-19-03892-t004:** Parameters for the simulation of an air flow inside a rectangular duct.

$v\left(y\right)\;[\frac{m}{s}]$	$v_{0}\;[\frac{m}{s}]$	$\rho_{f\;}[\frac{\mathrm{kg}}{m^{3}}]$	$\nu \;[\frac{m^{2}}{s}]$	$w\cdot h\;[m^{2}]$	Re[−]
see [44]	1.25	1.2	$1.5\times 10^{-5}$	0.02·0.04	≈2200

**Table 5 sensors-19-03892-t005:** Parameters for the simulation of a flow over an airfoil of a glider aircraft traveling at a height of 8000m adapted from [38] (example called Class 2). The sensor is located at $x_{0}$ on the airfoil with respect to front edge.

$v\left(y\right)\;[\frac{m}{s}]$	$v_{0}\;[\frac{m}{s}]$	$\rho_{f}\;[\frac{\mathrm{kg}}{m^{3}}]$	$\nu \;[\frac{m^{2}}{s}]$	$x_{0}\;[m]$	Re[−]
see (Equation 9), (Equation 10)	60	0.526	$2.9\times 10^{-5}$	0.2	$\approx 4.14\times 10^{5}$

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
