# Peer review of "An Artificial Vibrissa-Like Sensor for Detection of Flows [Author-notes fn1-sensors-19-03892]"

_sensors, 2019, doi:10.3390/s19183892_

Round 1
Reviewer 1 Report
The revised version of the manuscript is very well improved and the results presented support the claims made the authors. The manuscript certainly adds significant contribution to the literature of whisker-inspired sensor studies. The following minor modifications are suggested before the manuscript could be accepted for publication:
The way authors have introduced "remark 1, 2..." is not a common practice in journal manuscript writing. I recommend the authors to emphasize such remarks within the manuscript text without highlighting a separate statement under the title remark 1, 2... Seems there is an spell error in authors' names in reference 31 Reference 31 there is an error in the author The literature survey section must review the various whisker inspired sensors developed in the literature. Contributions by Kottapalli et al (Harbor seal whisker inspired flow sensors to reduce vortex-induced vibrations, doi: 10.1109/MEMSYS.2015.7051102) and Pablo valdivia et al (Design of a bio-inspired whisker sensor for underwater applications, doi: 10.1109/ICSENS.2012.6411517) should be reviewed.
Reviewer 2 Report
The authors have adequately dealt with all previous remarks. My recommendation is to accept the paper in its current form.
Author Response
Thanks for your first critical review and your efforts for the second round of review.
This manuscript is a resubmission of an earlier submission. The following is a list of the peer review reports and author responses from that submission.
Round 1
Reviewer 1 Report
In this work the authors present a theoretical model that includes large deflections of the whisker shaft which allows flow detection using support reaction at the base follicle.
The presentation of the concept the authors wanted to convey is poorly represented in the manuscript. The section on "experiment" presented in the manuscript is very unclear. the authors did not mention what the "sensor" actually is? Is this a real whisker? No schematics of the experimental setup are presented. The manuscript lacks any experimental evidence and the relevance of the scope of the submission to MDPI Sensors journal is very questionable.
Reviewer 2 Report
Scharff et al. propose the study of a flow sensor based on vibrissa-like structures. The paper is an extended version of the work published in the proceedings of the 18th International Conference on Mechatronics - Mechatronika (ME) (2018) and cited as reference [21]. The work is interesting and the manuscript is well organized. The experimental data are in good agreement with the simulation results. Nevertheless there are some concerns related to technical content that need to be addressed before the publication (see list below)
-) In the introduction, the authors should clarify better the novelties of their work with respect to the “state-of-the-art” and the differences from their previous work (ref. 21). They cite many papers about structures similar to the one they propose but they do not highlight the novel aspects of their work. Furthermore, they should clarify the advantages of the proposed device over flow sensors based on different transducing principles, for example thermal flow sensors and pressure difference flow sensors.
-) Figures 1 and 2 are incomprehensible without reading ref [8]. The author should add a figure (like Fig. 1 of ref [8]) in order to clarify the meaning of the symbols (A, B, C, D, E, alpha, beta, gamma, etc.) used in the figures.
-) In the experiment section, more details about the sensor shafts should be added, for example the material and the process used to fabricate them. A sketch or a photo of the shaft could be very useful. Furthermore, a sketch of the measurement set-up should be added in order to clarify better which are the measure conditions.
-) Figure 7: the authors should add a plot of the error between the calculated velocity and the measured one as a function of velocity in order to clarify the performance of the proposed flow sensor.
-) In the discussion section, the authors say that there are not information about the presence of a lift force in literature (lines 230-231). They should clarify better this sentence because there are works about dynamic lift force acting on cylinders (for a good review on this argument see, for example, C. Norberg “Fluctuating lift on a circular cylinder: review and new measurements” Journal of Fluids and Structures 17 (2003) 57–96).
-) The phrase “vibrissa of type B3” (page 3) is not clear.
-) The phrase “recorder for 30 s with 1 kHz” (line 146) is not clear.
-) The acronym w.r.t. should be removed.